# A Comprehensive View of the ASM1 Dynamic Model: Study on a Practical Case

**Carlos Costa**

Chemical Engineering Department, Faculty of Chemical Sciences, University of Salamanca, Plaza de la Merced s/n, 37008 Salamanca, Spain; ccosta@usal.es; Tel.: +34-923-294479; Fax: +34-923-294574

**Abstract:** The ASM1 model was elaborated by the IWA Task Group for Mathematical Modelling, with the aim of explaining and predicting the output values of organic matter concentration in activated sludge processes, especially for domestic wastewaters. In recent years, ASM1 has been completed with new components and extended to other biological processes, including biological membrane reactors, activated carbon filters, and microalgae bioreactors. In this article, the essentials of this model are studied by outlining the original topics that were formulated in the model, and by using a practical example of a wastewater treatment plant (WWTP), which can clarify the application of the ASM1. A protocol of approximation between the dynamic model and the experimental data for the COD effluent concentration is presented, based on three steps of tuning and fine tuning, and the corrected values of the kinetic parameters $Y_H$ and $\mu_{H,max}$ are calculated in accordance with the minimum error. In the simulation procedure, the baseline and dynamism are controlled, comparing them to the experimental data line, and the values obtained for the kinetic parameters are $Y_H = 0.60$ and $\mu_{H,max} = 0.40 \text{ d}^{-1}$. The kinetic parameters reflect the activity of the mixed community of microorganisms in the WWTP.

**Keywords:** biological wastewater treatment; dynamic model; activated sludge; ASM

## 1. Introduction

Biological wastewater treatment is strongly influenced by environmental perturbations. Variations in temperature, influent organic matter, biomass concentration, and influent flow modify the bacterial community activity and produce a modification in the output concentration of the organic matter. Traditional stationary mathematical models do not consider this situation, which occurs daily in wastewater treatment plants (WWTP).

A dynamic model is a non-stationary mathematical model, which considers the dynamic behaviour of the variables affecting the output parameters. The IWA Task Group for Mathematical Modelling (International Water Association), conducted by Morgens Henze, published the first activated sludge model (ASM1) [1] in 1987, which is a deterministic model (based on basic engineering principles), initially formulated as a general mass balance to the biological reactor. This original model was extended to phosphorus and nitrogen removal, oxygen consumption, and sludge production [2–4]. All ASMs are recorded in the IWA Report No. 9 [5].

The ASM1 includes 13 components (different types of substrates and biomass) and 8 fundamental processes (growth and substrate utilisation rates), resulting, initially, in 13 mass balance equations, with almost 20 kinetic parameters. The model was initially proposed for domestic wastewater, and was validated more than a decade ago in its application to specific cases of WWTPs [6,7]. The ASM1 is often too complicated for solving predictions in real-scale operations at wastewater treatment plants. The characterisation of wastewater for mathematical modelling with the ASM1 can sometimes be hard and time consuming. In addition, the use of so many parameters for the model can result in poor parameter identifiability [8].

The ASM1 was designed to be adapted to the special characteristics of the wastewater and the biological treatments applied. In other words, the system of differential equations is limited to the components in consideration, and the process rates are selected as a function of the biological treatment conditions.

In recent years, the ASM1 has been improved by including new components or processes to the original model [9,10], because of the special and flexible configuration of the ASM1, which permits continuous updates and adjustments. The application of this dynamic model is not only possible for activated sludge processes [11–14], but also for other biological treatments, such as membrane bioreactors [15,16] and microalgae systems [17]. In fact, the ASM1 represents the basis for the formulation of new mathematical models for nutrient removal in WWTPs [18] and in granular activated carbon filters [19], and for the application of activated sludge models (ASMs) to MBR systems [20,21].

The ASM1 has been applied to diverse biological processes, for suspension growth or fixed growth, for bacteria or microalgae, but the detailed procedure of formulating the model, related to the election of components and processes, and especially to the protocol of simulation, has not been described in depth, and needs an explanation for more extended use, mainly in WWTPs. Predicting the COD value in a WWTP under abnormal situations or special environmental events is of great value for adequate correction of the effluents, and for preserving the quality of natural streams.

In the present work, the ASM1 will be described in depth, and, as a novelty, a protocol of simulation, based on the approximation between the dynamic model and real data (tuning), will be proposed. A simulation of the output COD concentration will be performed from a medium–high-sized WWTP (260,000 habitants), considering dynamism in the reactor temperature (affecting $\mu_H$), biomass concentration, and influent COD. The effect of dynamic variables on the output parameter (COD) will be analysed from the biological treatment point of view, explaining the behaviour of the bacterial community, focusing on the response to organic matter removal.

## 2. Materials and Methods

### 2.1. Dynamic Model ASM1

The mathematical expression for the generic mass balance applied to the vector $\xi$, representing substrate or biomass concentration, is as follows (Figure 1):

$$V\frac{d\xi}{dt} = Q\xi_{in} - Q\xi + Vr(\xi) \tag{1}$$

$$\frac{d\xi}{dt} = r(\xi) + \frac{1}{\Theta}(\xi_{in} - \xi) \tag{2}$$

where $\Theta$ in Equation (2) is the hydraulic residence time (HRT) and $r(\xi)$ is the conversion vector of the variable $\xi$ (substrate utilisation global rate).

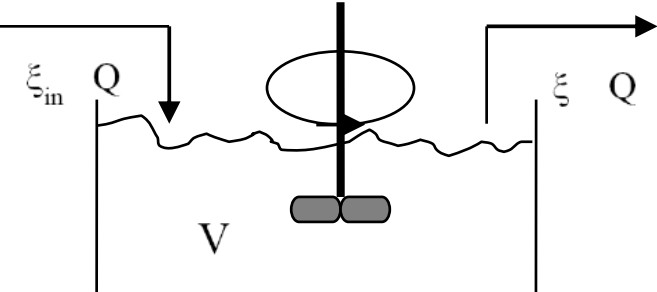

**Figure 1.** CSTR scheme for the ASM1. $\xi$ is the vector of reactor and effluent concentration, $\xi_{in}$ is the vector of influent concentration, $Q$ is the influent flow rate, and $V$ is the reactor volume.

The substrate utilisation global rate ($r_i$) in the ASM1 is a conversion rate for the component $i$ by the process $j$, as follows:

$$r_i = \sum_j v_{ij} \rho_j \tag{3}$$

where $v_{ij}$ is the stoichiometric coefficient and $\rho_j$ is the process rate. This equation defines a differential equation system, in which every differential equation has the form of Equation (2) for each component $i$ of the model, affected by the different process rates designed by the $j$ subscript. The process rates that affect the components of the ASM1 are described in the Peterson matrix (Table 1).

Two of the most remarkable and valuable points of the ASM1 are the division of the components into fractions and the separation of the rates into processes. Organic matter is divided into soluble and rapidly biodegradable ($S_s$), and particulate and slowly biodegradable ($X_s$), so the notation $S$ means soluble, $X$ means particulate, and the subscripts refer to the nature of the substrate. The most important processes affecting organic matter removal are the aerobic and anaerobic growth of heterotrphs, and the decay and hydrolysis of particulate organic matter (Table 2).

### 2.2. Uncoupling of the Components

When applying the ASM1 for the simulation of oscillation in the output organic matter concentration in a WWTP, 3 differential equations must be written for soluble and particulate substrates ($S_s$ and $X_s$), and for heterotrophic biomass ($X_{BH}$). The system of these equations for the 3 components (numbers 2, 4 and 5 in Table 1), in accordance with the process rates that affect them, is as follows:

$$\frac{dS_s}{dt} = -\frac{\mu_H}{Y_H}\left(\frac{S_s}{K_s+S_s}\right)\left(\left(\frac{S_o}{K_{OH}+S_o}\right) + \eta_g\left(\frac{K_{OH}}{K_{OH}+S_o}\right)\left(\frac{S_{NO}}{K_{NO}+S_{NO}}\right)\right)X_{BH} + k_h\left(\frac{X_{BH}}{K_x X_{BH}+X_s}\right)\left(\left(\frac{S_o}{K_{OH}+S_o}\right) + \eta_h\left(\frac{K_{OH}}{K_{OH}+S_o}\right)\left(\frac{S_{NO}}{K_{NO}+S_{NO}}\right)\right)X_s + \frac{1}{\Theta}(S_{s,in} - S_s) \tag{4}$$

$$\frac{dX_s}{dt} = \left(1-f_p\right)b_H X_{BH} + \left(1-f_p\right)b_A X_{BA} - k_h\left(\frac{X_{BH}}{K_x X_{BH}+X_s}\right)\left(\left(\frac{S_o}{K_{OH}+S_o}\right) + \eta_h\left(\frac{K_{OH}}{K_{OH}+S_o}\right)\left(\frac{S_{NO}}{K_{NO}+S_{NO}}\right)\right)X_s + \frac{1}{\Theta}(X_{s,in} - X_s) \tag{5}$$

$$\frac{dX_{BH}}{dt} = \mu_H\left(\frac{S_s}{K_s+S_s}\right)\left(\left(\frac{S_o}{K_{OH}+S_o}\right) + \eta_g\left(\frac{K_{OH}}{K_{OH}+S_o}\right)\left(\frac{S_{NO}}{K_{NO}+S_{NO}}\right)\right)X_{BH} - b_H X_{BH} + \frac{1}{\Theta}(X_{BH,in} - X_{BH}) \tag{6}$$

The complexity of solving these differential equations is evident and highly incremented because variables are coupled among different equations. For solving differential equations separately, coupled variables present in mass balance equations can be introduced in the model as analysed parameters (discrete values). In this case, mass balance equations become independent and can be solved separately. For example, the solution of Equation (4) for soluble substrates predicts output COD values in biological treatments. This procedure, in which numerical values of the dynamic variables are introduced for the solution of differential equations, is named "uncoupling".

### 2.3. Approximations in Process Rates

Equations (4)–(6) need to be approximated in order to reduce unnecessary complexity and for good parameter identifiability, because having so many parameters leads to the loss of a good response in the output variables. These approximations are in accordance with the nature of wastewater and the operating conditions of the biological process.

Normally, in WWTPs, wastewater entering the biological process after primary treatment has a relatively low concentration of solids, so hydrolysis is not a predominant process in substrate reduction. In addition, oxygen concentration is known to be not limiting over 1.5 mg/L, and in biological processes, it is maintained over this value.

**Table 1.** Peterson matrix for the ASM1. Components are recorded in columns and processes can be identified in rows.

| i—Component→ <br> j—Process↓ | 1 <br> $S_I$ | 2 <br> $S_S$ | 3 <br> $X_I$ | 4 <br> $X_S$ | 5 <br> $X_{BH}$ | 6 <br> $X_{BA}$ | 7 <br> $X_P$ | 8 <br> $S_O$ | 9 <br> $S_{NO}$ | 10 <br> $S_{NH}$ | 11 <br> $S_{ND}$ | 12 <br> $X_{ND}$ | 13 <br> $S_{ALK}$ |
|---|---|---|---|---|---|---|---|---|---|---|---|---|---|
| 1-Aerobic growth of heterotrophs | | $-\frac{1}{Y_H}$ | | | 1 | | | $-\frac{1-Y_H}{Y_H}$ | | $-i_{XB}$ | | | $\frac{-i_{XB}}{14}$ |
| 2-Anoxic growth of heterotrophs | | $-\frac{1}{Y_H}$ | | | 1 | | | | $-\frac{1-Y_H}{2.86Y_H}$ | $-i_{XB}$ | | | $\frac{1-Y_H}{14\times2.86Y_H}-\frac{i_{XB}}{14}$ |
| 3-Aerobic growth of autotrophs | | | | | | 1 | | $-\frac{4.57-Y_A}{Y_A}$ | $\frac{1}{Y_A}$ | $-i_{XB}-\frac{1}{Y_A}$ | | | $\frac{-i_{XB}}{14}-\frac{1}{7Y_A}$ |
| 4-Decay of heterotrophs | | | | $1-f_P$ | $-1$ | | $f_P$ | | | | | $-i_{XB}-f_P i_{XP}$ | |
| 5- Decay of autotrophs | | | | $1-f_P$ | | $-1$ | $f_P$ | | | | | $-i_{XB}-f_P i_{XP}$ | |
| 6-Ammonification of soluble organic nitrogen | | | | | | | | | | 1 | $-1$ | | $\frac{1}{14}$ |
| 7-Hydrolysis of entrapped organics | | 1 | | $-1$ | | | | | | | | | |
| 8-Hydrolysis of entrapped organic nitrogen | | | | | | | | | | | 1 | $-1$ | |

**Table 2.** Expression of the process rates for ASM1 in accordance with Henze et al. (2000). Numbers of the processes are identified by *j* subscript.

| Process Rate | Mathematical Expression |
|---|---|
| 1—Heterotrophs, aerobic growth | $\mu_H \left( \frac{S_s}{K_s + S_s} \right) \left( \frac{S_o}{K_{OH} + S_o} \right) X_{BH}$ |
| 2—Heterotrophs, anaerobic growth | $\mu_H \left( \frac{S_s}{K_s + S_s} \right) \left( \frac{K_{OH}}{K_{OH} + S_o} \right) \left( \frac{S_{NO}}{K_{NO} + S_{NO}} \right) \eta_g X_{BH}$ |
| 3—Autotrophs, aerobic growth | $\mu_A \left( \frac{S_{NH}}{K_{NH} + S_{NH}} \right) \left( \frac{S_o}{K_{OA} + S_O} \right) X_{BA}$ |
| 4—Heterotrophs, decay | $b_H X_{BH}$ |
| 5—Autotrophs, decay | $b_A X_{BA}$ |
| 6—Organic nitrogen, ammonification | $k_a S_{ND} X_{BH}$ |
| 7—Hydrolysis of particulate organic matter | $k_h \left( \frac{X_s/X_{BH}}{K_X + X_s/X_{BH}} \right) \left( \left( \frac{S_o}{K_{OH} + S_o} \right) + \eta_h \left( \frac{K_{OH}}{K_{OH} + S_o} \right) \left( \frac{S_{NO}}{K_{NO} + S_{NO}} \right) \right) X_{BH}$ |
| 8—Hydrolysis of particulate organic nitrogen | $k_h \left( \frac{X_s/X_{BH}}{K_X + X_s/X_{BH}} \right) \left( \left( \frac{S_o}{K_{OH} + S_o} \right) + \eta_h \left( \frac{K_{OH}}{K_{OH} + S_o} \right) \left( \frac{S_{NO}}{K_{NO} + S_{NO}} \right) \right) X_{BH} \left( \frac{X_{ND}}{X_s} \right)$ |

If these approximations are assumed, the process rates of the hydrolysis of particulate organic matter (Table 2) are not considered in Equations (4) and (5), anoxic growth of heterotrophs does not affect Equations (4) and (6), and oxygen concentration (Monod term) in aerobic growth of heterotrophs is not limiting and, in consequence, is not considered in Equations (4)–(6).

With these approximations, the system of these 3 differential equations is as follows:
Soluble substrate:

$$\frac{dS_s}{dt} = -\frac{\mu_H}{Y_H} \left( \frac{S_s}{K_s + S_s} \right) X_{BH} + \frac{1}{\Theta} (S_{s,in} - S_s) \tag{7}$$

Particulate substrate:

$$\frac{dX_s}{dt} = (1 - f_p) b_H X_{BH} + (1 - f_p) b_A X_{BA} + \frac{1}{\Theta} (X_{s,in} - X_s) \tag{8}$$

Heterotrophic biomass:

$$\frac{dX_{BH}}{dt} = \mu_H \left( \frac{S_s}{K_s + S_s} \right) X_{BH} - b_H X_{BH} + \frac{1}{\Theta} (X_{BH,in} - X_{BH}) \tag{9}$$

At this point, we must define which are constants in the model in time (kinetic parameters) and which are dynamic variables, modifying in time and introduced as routine, time-dependent numerical values in the differential equation to be solved. A mathematical program must be elaborated for this purpose. Because of the use of discrete values for the dynamic variables, which are present in several equations ($X_{BH}$ and $\mu_H$), differential equations can be solved separately (uncoupling).

### 2.4. Experimental Setup

Equation (7) is the differential equation in which the simulation of effluent COD was obtained. Dynamic variables that affect the output COD value are biomass concentration ($X_{BH}$) and specific growth rate ($\mu_H$); these are both influenced by temperature ($\mu_H = \mu_{H,max} 1.072^{(T-20)}$). $S_{s,in}$ is also included as a dynamic variable because of daily fluctuations in the WWTP. Equation (7) was solved by MATLAB R2021b [6,22], adjusting the kinetic parameters $Y_H$ and $K_s$, using the known range 0.40–0.75 for $Y_H$ and using the medium value of effluent COD for $K_s$ [23].

The protocol of simulation is separated into three steps for the approximation of the simulation line to the experimental data. In the first step (tuning on $Y_H$), the value of $Y_H$ is obtained, in the range 0.5–0.7 for $\mu_{H,max} = 0.1$ d$^{-1}$ as a fixed value. This value of the maximum specific growth rate is low for better visualisation of the dynamism of the simulation. In the second step (tuning on $\mu_{H,max}$), the value of $\mu_{H,max}$ is selected maintaining the fixed value for $Y_H$ obtained in the first step. In the third step (fine tuning on $\mu_{H,max}$),

the value of $\mu_{H,max}$ is carefully adjusted, comparing errors between the simulation and the experimental values.

### 2.5. Biological Treatment in WWTP

The wastewater treatment plant used in this study is a medium–high-sized plant, designed for 260,000 habitants of the city of Salamanca (Spain). The scheme of the treatment processes is described in Figure 2.

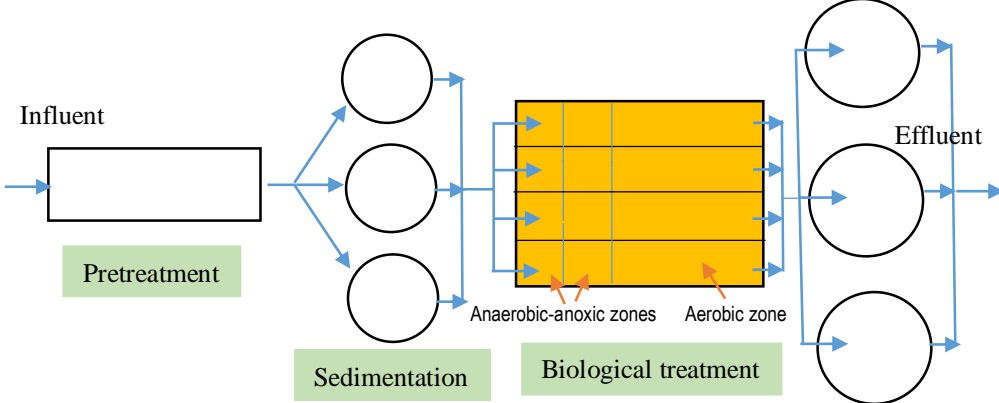

**Figure 2.** Scheme of the WWTP. Average influent flow rate was 60,466 m$^3$/d during the measuring time. Total biological reactor volume (orange colour) was 37,240 m$^3$, 4 chambers of 9310 m$^3$ ($\Theta$ = 14.8 h), and oxygen concentration was maintained in 1.5 $\pm$ 0.3 mg/L.

### 2.6. Analytical Methods

COD was analysed as total COD in influent and effluent of the biological treatment, following the standard method [24]. Biomass concentration in the biological reactor was determined as MLSS (mixed liquor suspended solids), which is the solid residue after evaporation at 103–105 °C in samples filtered by filters with a pore size of 0.45 µm [24]. The temperature was measured in the biological reactor by a thermocouple.

## 3. Results and Discussion

### 3.1. Dynamic Variables

The simulation of effluent COD in the biological treatment of a WWTP relies on the solution of Equation (7), in which $X_{BH}$, $\mu_H$ and $S_{s,in}$ are the three dynamic variables included in the MATLAB programme, as time-dependent, discrete values (analysed values). These discrete values were obtained from Table 3.

**Table 3.** Experimental values for biological treatment measured in the WWTP between May 2020 and April 2021 (343 days). Data were supplied by the staff of the WWTP.

| Day | COD$_{in}$ (mg/L) | COD (mg/L) | Temp (°C) | Biomass (mg/L) |
|---|---|---|---|---|
| 1 | —— | 23.7 | 17.6 | 2723 |
| 5 | 367 | 32.9 | 18.3 | 2160 |
| 7 | 302 | 27.3 | 18.5 | 2210 |
| 12 | 229 | 25.3 | 18.3 | 1967 |
| 14 | 250 | 24.1 | 18.9 | 1850 |
| 19 | 290 | 21.6 | 19.2 | 1477 |
| 21 | 338 | 21.0 | 18.7 | 1730 |
| 26 | 331 | 27.2 | 18.4 | 1623 |
| 29 | 365 | 34.1 | 19.0 | 1720 |
| 33 | 326 | 29.3 | 19.5 | 1367 |
| 35 | 369 | 31.1 | 19.8 | 1433 |
| 40 | 342 | 25.9 | 20.5 | 1790 |

**Table 3.** *Cont.*

| Day | COD$_{in}$ | COD | Temp | Biomass |
| --- | --- | --- | --- | --- |
| | (mg/L) | (mg/L) | (°C) | (mg/L) |
| 42 | 354 | 25.6 | 20.8 | 1787 |
| 47 | 290 | 23.7 | 21.1 | 1610 |
| 49 | 342 | 23.9 | 21.6 | 1647 |
| 54 | 291 | 26.2 | 21.4 | 1470 |
| 56 | 331 | 23.7 | 22.5 | 1447 |
| 61 | 269 | 18.6 | 21.7 | 1677 |
| 63 | 324 | 24.3 | 21.8 | 1277 |
| 68 | 284 | 45.6 | 22.0 | 1210 |
| 70 | 251 | 22.4 | 22.8 | 1310 |
| 77 | 364 | 31.4 | 22.4 | 810 |
| 82 | 478 | 25.9 | 22.9 | 2103 |
| 84 | 321 | 40.6 | 22.8 | 2263 |
| 89 | 265 | 21.5 | 21.6 | 1993 |
| 91 | 308 | 21.5 | 21.8 | 1770 |
| 96 | 278 | 21.1 | 21.8 | 1357 |
| 98 | 280 | 19.3 | 22.1 | 917 |
| 105 | 350 | 26.7 | 21.6 | 1790 |
| 113 | 321 | 27.5 | 21.1 | 1227 |
| 118 | 303 | 29.9 | 22.1 | 1410 |
| 120 | 322 | 35.0 | 21.4 | 1177 |
| 125 | 296 | 17.8 | 21.1 | 1527 |
| 127 | 267 | 22.7 | 21.1 | 1480 |
| 132 | 373 | 37.0 | 21.0 | 1673 |
| 134 | 368 | 30.2 | 20.3 | 1300 |
| 135 | 347 | 26.7 | 21.5 | 2027 |
| 139 | 384 | 26.9 | 19.5 | 1945 |
| 141 | 326 | 25.0 | 19.7 | 1715 |
| 147 | 390 | 27.2 | 19.8 | 1560 |
| 149 | 382 | 28.3 | 18.8 | 1623 |
| 153 | 408 | 23.6 | 18.8 | 1485 |
| 155 | 189 | 15.4 | 16.5 | 1895 |
| 160 | 244 | 18.1 | 17.4 | 1650 |
| 162 | 306 | 20.2 | 18.1 | 1968 |
| 167 | 352 | 27.6 | 18.1 | 1553 |
| 169 | 332 | 33.2 | 17.8 | 1618 |
| 173 | 198 | 23.2 | 14.9 | 1675 |
| 175 | 311 | 25.8 | 17.5 | 1635 |
| 180 | 300 | 33.9 | 17.2 | 1608 |
| 182 | 372 | 25.9 | 17.4 | 1490 |
| 187 | 369 | 24.7 | 17.0 | 2590 |
| 189 | 361 | 17.7 | 17.0 | 2215 |
| 194 | 392 | 27.3 | 16.5 | 2590 |
| 195 | 365 | 24.1 | 16.1 | 1895 |
| 197 | 469 | 12.6 | 15.7 | 2105 |
| 203 | 347 | 18.6 | 14.0 | 2255 |
| 208 | 349 | 24.9 | 15.4 | 1665 |
| 210 | 306 | 23.5 | 14.9 | 1860 |
| 215 | 340 | 29.5 | 15.1 | 2008 |
| 222 | 341 | 26.9 | 11.4 | 1475 |
| 229 | 379 | 37.3 | 13.5 | 1748 |
| 232 | 427 | 38.9 | 13.4 | 2045 |
| 236 | 396 | 32.1 | 12.2 | 2690 |
| 239 | 372 | 30.5 | 11.9 | 2483 |
| 243 | 384 | 30.7 | 12.1 | 2525 |
| 246 | 259 | 23.5 | 11.1 | 2778 |
| 250 | 277 | 53.2 | 12.5 | 2283 |
| 253 | 390 | 58.1 | 13.2 | 2325 |

**Table 3.** *Cont.*

| Day | COD$_{in}$ | COD | Temp | Biomass |
|---|---|---|---|---|
| | (mg/L) | (mg/L) | (°C) | (mg/L) |
| 257 | 358 | 49.5 | 13.3 | 1948 |
| 260 | 114 | 28.1 | 13.8 | 2133 |
| 264 | 196 | 33.3 | 9.1 | 1808 |
| 267 | 215 | 24.0 | 11.6 | 3013 |
| 271 | 251 | 20.3 | 11.8 | 4375 |
| 274 | 253 | 24.8 | 13.0 | 3510 |
| 278 | 259 | 20.7 | 12.5 | 3165 |
| 281 | 264 | 27.4 | 13.1 | 2533 |
| 285 | 275 | 32.1 | 12.6 | 2575 |
| 288 | 134 | 23.6 | 13.8 | 2343 |
| 292 | 304 | 24.1 | 14.0 | 2205 |
| 295 | 300 | 26.8 | 14.1 | 1983 |
| 299 | 319 | 28.8 | 14.1 | 1838 |
| 301 | 284 | 24.3 | 13.9 | 1855 |
| 306 | 334 | 26.9 | 13.5 | 2260 |
| 309 | 323 | 47.6 | 14.8 | 1913 |
| 313 | 352 | 28.6 | 14.1 | 2008 |
| 320 | 329 | 30.8 | 14.7 | 2530 |
| 323 | 301 | 27.1 | 15.1 | 2340 |
| 327 | 287 | 22.7 | 15.1 | 2190 |
| 334 | 310 | 24.5 | 15.2 | 2050 |
| 336 | 283 | 27.4 | 15.9 | 2060 |
| 341 | 254 | 62.1 | 15.8 | 1830 |
| 343 | 235 | 19.9 | 15.7 | 1090 |

*3.2. Tuning on $Y_H$*

Tuning is the operation in which the simulation is adjusted to make it coincide with the experimental data. The first time that the dynamic model response is checked, the behaviour of $Y_H$ should be assayed. The initial value of $\mu_{H,max}$ normally marks the baseline of the simulation, and, in this first approximation, it was decided that $\mu_{H,max}$ would be low for better visualisation of the mathematical response to the output of COD and the dynamic variables, because the dynamism was expected to be reduced, increasing the $\mu_{H,max}$ value. The value selected was $\mu_{H,max} = 0.1$ d$^{-1}$ (Figure 3), a low value compared with the ranges for domestic wastewater in the literature: 0.45–1.0 d$^{-1}$ [16], 0.6–13.2 d$^{-1}$ [23], and 3.0–13.2 d$^{-1}$ for the original ASM1 [1] were proposed. For tuning on $Y_H$, $K_s = 27.7$ mg/L (average value of effluent COD) and $\Theta = 14.8$ h (biological process of WWTP in Figure 2).

The main conclusion (Figure 3) in the prediction of the output $S_s$ value in the WWTP is that the higher the fraction of substrate incorporated into the biomass, the higher the concentration of remaining organic matter after the biological treatment ($Y_H = 0.7$ elevates the simulation line).

The elevation of the simulation line when the $Y_H$ value was incremented can be explained by the fact that the formation of the biomass is a slow process (synthesis reaction), compared to the transformation of organic matter (oxidation reaction).

Mathematically, increasing the $Y_H$ value in Equation (7) makes the derivative of $S_s$ less negative, because the negative term normally has a higher value in the equation, leading to a smaller decrease from the initial value (output substrate value rises).

The value assumed for $Y_H$ in the literature is normally 0.6 [5,11,12], and this is the fixed value used in subsequent tuning assays.

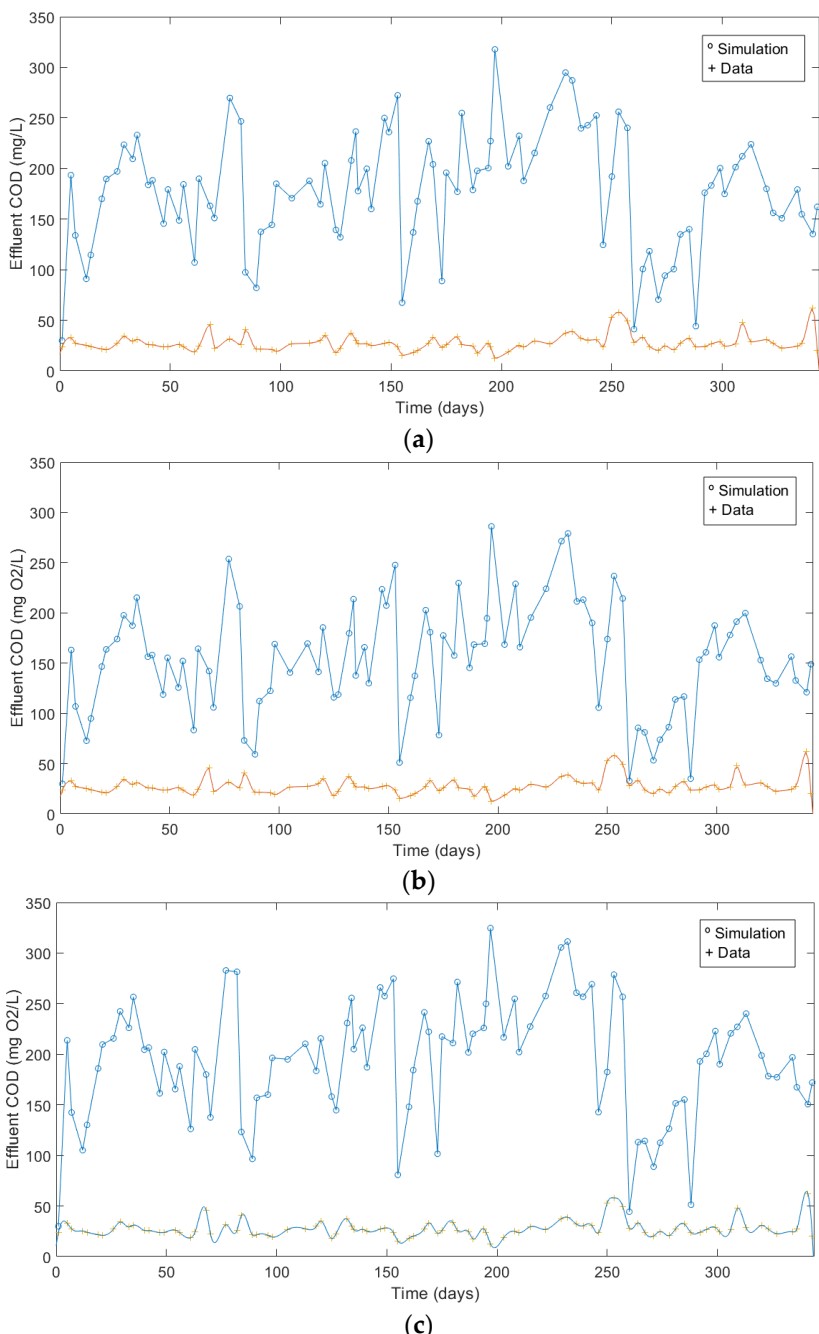

**Figure 3.** Tuning on $Y_H$ for $\mu_{H,max} = 0.1 \text{ d}^{-1}$. (**a**) $Y_H = 0.6$, (**b**) $Y_H = 0.5$ and (**c**) $Y_H = 0.7$. Blue upper line is simulation data and coloured lower line is the evolution of real values (analysed data) after biological treatment in the WWTP (Table 3, COD (mg/L)).

### 3.3. Tuning on $\mu_{H,max}$

Specific growth rates are strongly affected by temperature in biological processes. The mathematical expression of this influence is $\mu_H = \mu_{H,max} \, 1.072^{(T-20)}$, and modification of the $\mu_{H,max}$ value will generate an inverse response on $S_s$. Increments in $\mu_{H,max}$ will decrease the effluent organic matter concentration ($S_s$), because the substrate will be degraded to a greater extent.

Figure 4 shows two situations, in which the simulation results in an underestimation (a), where the simulation line predicts lower values of organic matter concentration ($\mu_{H,max} = 0.6 \text{ d}^{-1}$), and in an overestimation (b), with the simulation line over the line of

experimental data ($\mu_{H,max} = 0.3$ d$^{-1}$). The simulation will be coincident for a value of $\mu_{H,max}$ in between those values.

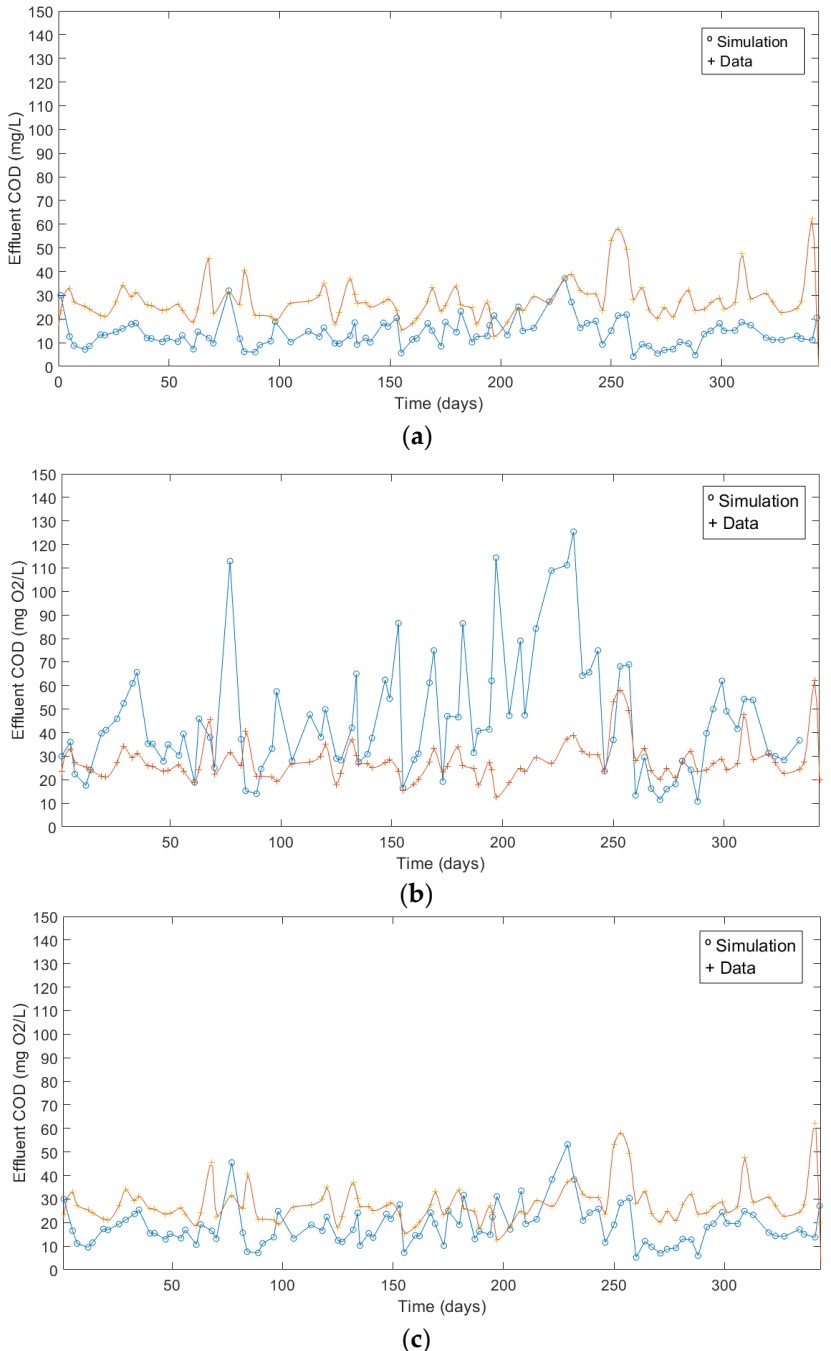

**Figure 4.** Tuning on $\mu_{H,max}$ for $Y_H = 0.6$. (**a**) $\mu_{H,max} = 0.6$ d$^{-1}$, (**b**) $\mu_{H,max} = 0.3$ d$^{-1}$ and (**c**) $\mu_{H,max} = 0.5$ d$^{-1}$. Simulation is the blue line and analysed data is the orange line.

On the other hand, comparing the graphs in Figure 4, increments in the $\mu_{H,max}$ value lowered the dynamism in the dynamic model simulation (Figure 4a,c), and fluctuations in the output value were higher when $\mu_{H,max}$ was lower (Figure 4b). Increasing the $\mu_{H,max}$ value tempers dynamism because the simulation reduced the fluctuations of the output parameter (more visible for a higher reduction in substrate), which means higher actuation of the dynamic model in Equation (7) [25,26].

### 3.4. Fine Tuning on $\mu_{H,max}$

When the simulation is approximated to the analysed data (Figure 4c), quantification of fitting (fine tuning) in the dynamic model, visualised in Figure 5, has to be performed. In Table S1 (Supplementary Material), the average error between the simulated and analysed data is presented, after the COD and simulated values in the table. In this case, the fine tuning in Figure 5 was solved for $\mu_{H,max}$ values close to 0.40 d$^{-1}$ (0.38 and 0.42), with $Y_H$ = 0.60 as a fixed value. The error is calculated by considering the analysed value as the true value, as follows:

$$Error = \frac{(simulated\ value - analysed\ value)}{analysed\ value} \times 100$$

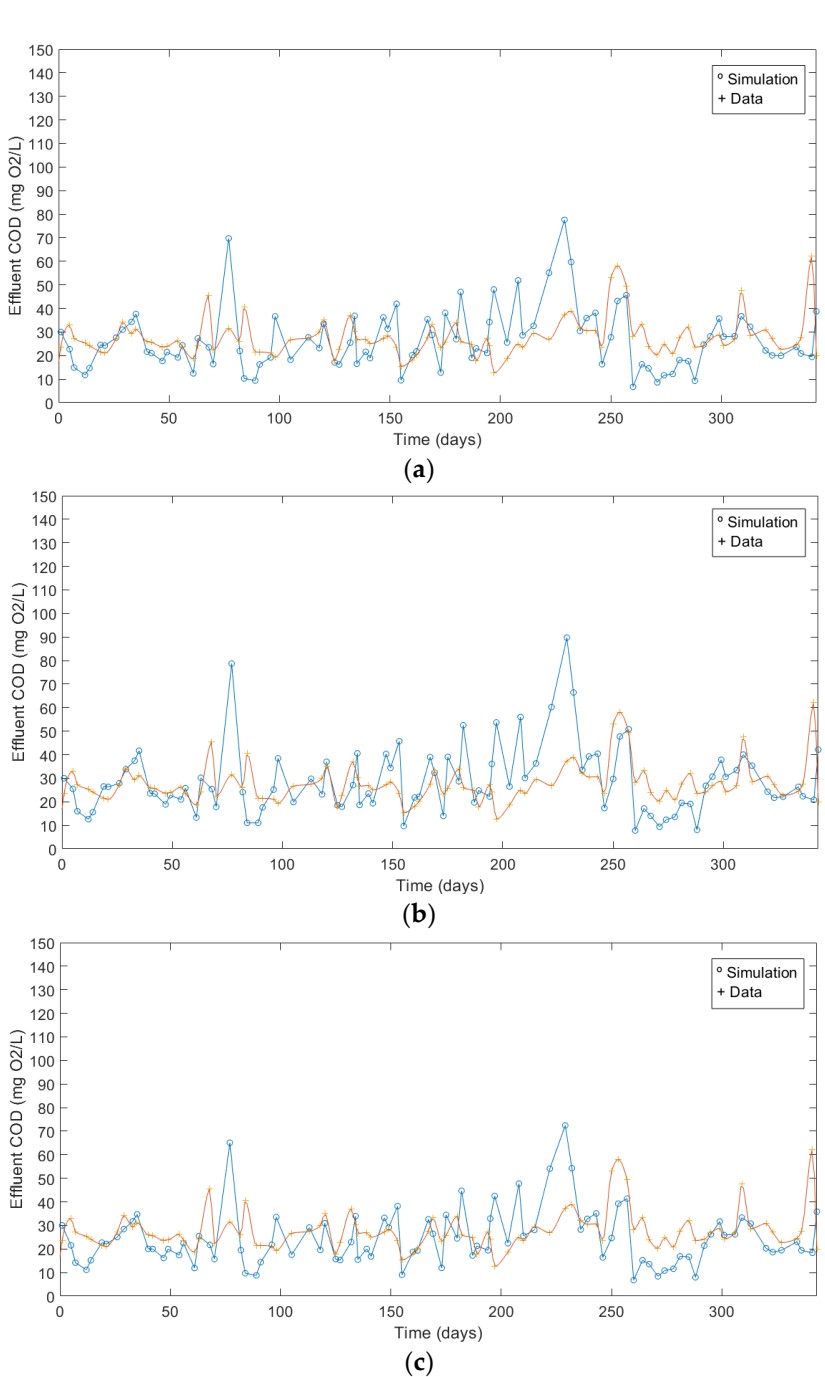

**Figure 5.** Fine tuning on $\mu_H$ for $Y_H$ = 0.60. (**a**) $\mu_{H,max}$ = 0.40 d$^{-1}$, (**b**) $\mu_{H,max}$ = 0.38 d$^{-1}$ and (**c**) $\mu_{H,max}$ = 0.42 d$^{-1}$.

The average errors show an overestimation for $\mu_{H,max}$ = 0.38 d$^{-1}$ (error = 8.1%) and an underestimation for $\mu_{H,max}$ = 0.42 d$^{-1}$ (error = −7.7%). The minimum error between the simulated and analysed values (COD (mg/L)) was obtained for $\mu_{H,max}$ = 0.40 d$^{-1}$ (error = −0.5%), which was the value selected for the maximum specific growth rate of the community of microorganisms in the studied biological treatment of the WWTP. This value of $\mu_{H,max}$ is very close to the range 0.45–1.0 d$^{-1}$ proposed by other authors [16,27]; although, in other articles, the proposed value is higher [28].

In Table S1 (Supplementary Material), high values of individual relative errors can be observed (as visualised in Figure 5). This is a normal result in dynamic simulations of biological treatments [15], in which the consequences of a "live" system are often observed. The main reasons for overestimation and underestimation of the dynamic mathematical model are especially related to the activity of biomass (active and inert fractions), attenuation and inertial effect of the bacterial community on fluctuations in temperature, the flow rate, and the substrate concentration [29,30]. A dynamic mathematical model does not reproduce this behaviour of the bacterial community, no theoretical model is able to do this, but its utility, especially for the prediction of the response to perturbations and the oscillatory behaviour of the output parameter, is evident [31].

## 4. Conclusions

The ASM1 is the basis for the mathematical modelling of organic matter reduction in aerobic biological wastewater treatment systems. Although the original model was published more than 30 years ago, the configuration and flexibility of this model make it almost universal in the explanation of biological treatment processes.

Understanding how it was formulated and how it works is of great value for the mathematical modelling of real systems, and for new proposals in the modelling of special systems. One of the most important steps in ASM1 use is the selection of the processes involved after the components are selected. Adjustment and simplification of the dynamic model are crucial for the correct application of this useful model.

When applying ASM1 dynamic modelling for the prediction of a WWTP, tuning on $Y_H$ and $\mu_{H,max}$ has to be elaborated. In the case of the WWTP of Salamanca (Spain, 260,000 habitants), $Ks$ is fixed as the average value of effluent COD ($Ks$ = 27.7 mg/L), and $\Theta$ is in accordance with the reactor volume and the influent flow rate ($\Theta$ = 14.8 h). Assuming a short range of $Y_H$ values (0.40–0.75) in the protocol of simulation proposed in this article, in which the approximation of the simulation line to the experimental data is performed in three steps, tuning on $Y_H$, conducts to $Y_H$ = 0.60, and tuning on $\mu_{H,max}$, after the approximation of the baseline and dynamism, comparative errors between the simulated and analysed data mark the correct value (fine tuning, $\mu_{H,max}$ = 0.40 d$^{-1}$).

Individual errors are high in the dynamic modelling of biological treatments; a live system reproduced by a mathematical model will inevitably lead to this result, but utility in the prediction of the oscillatory behaviour of the output parameter value, especially when environmental perturbations occur, is essential.

**Supplementary Materials:** The following supporting information can be downloaded at: https://www.mdpi.com/article/10.3390/w14071046/s1, Table S1: Table of errors between simulated values (Sim) and analysed values (COD) for fine tuning, $Y_H$ = 0.60 and $\mu_{H,max}$ = 0.40, 0.38 and 0.42 d$^{-1}$.

**Funding:** This research received no external funding.

**Institutional Review Board Statement:** Not applicable.

**Informed Consent Statement:** Not applicable.

**Acknowledgments:** The author expresses special thanks and their appreciation to Teodoro García (Plant Manager) and the staff of the Wastewater Treatment Plant of Salamanca for the information supplied about the characteristics of the WWTP, analysed data and their contribution to this work.

**Conflicts of Interest:** The author declares no conflict of interest.

## Abbreviations

Nomenclature

| | |
|---|---|
| $b_A$ | decay coefficient for autotrophic biomass ($d^{-1}$); |
| $b_H$ | decay coefficient for heterotrophic biomass ($d^{-1}$); |
| $f_p$ | fraction of biomass leading to particulate products; |
| $i_{XB}$ | nitrogen fraction in biomass; |
| $i_{XP}$ | nitrogen fraction in products from biomass; |
| $k$ | kinetic coefficient ($d^{-1}$); |
| $k_h$ | hydrolysis rate constant ($d^{-1}$); |
| $K_{OH}$ | oxygen half-saturation coefficient for heterotrophic biomass (mg/L); |
| $K_s$ | half-saturation coefficient for readily biodegradable substrate (mg/L); |
| $K_X$ | half-saturation coefficient for particulate biodegradable substrate (mg/L); |
| $Q$ | influent flow rate (L/d); |
| $r_i$ | substrate utilization rate (mg/(L d)); |
| $r(\xi)$ | conversion vector of the variable $\xi$ (mg/(L d)); |
| $S_{ALK}$ | alkalinity (mol/L); |
| $S_I$ | soluble inert organic matter (mg/L); |
| $S_{ND}$ | soluble biodegradable organic nitrogen (mg/L); |
| $S_{NH}$ | ammonia nitrogen (mg/L); |
| $S_{NO}$ | nitrate and nitrite nitrogen (mg/L); |
| $S_O$ | dissolved oxygen (mg/L); |
| $S_S$ | readily biodegradable substrate (mg/L); |
| $S_{S,in}$ | influent readily biodegradable substrate (mg/L); |
| $t$ | time (d); |
| $T$ | temperature (°C); |
| $V$ | reactor volume (L); |
| $X_{BA}$ | active autotrophic biomass (mg/L); |
| $X_{BH}$ | active heterotrophic biomass (mg/L); |
| $X_{BH,in}$ | influent active heterotrophic biomass (mg/L); |
| $X_I$ | particulate inert organic matter (mg/L); |
| $X_{ND}$ | particulate biodegradable organic nitrogen (mg/L); |
| $X_P$ | particulate products arising from biomass decay (mg/L); |
| $X_S$ | slowly biodegradable substrate (mg/L); |
| $X_{S,in}$ | influent slowly biodegradable substrate (mg/L); |
| $Y_A$ | growth yield of autotrophic biomass; |
| $Y_H$ | growth yield of heterotrophic biomass. |

Greek symbols

| | |
|---|---|
| $\xi$ | vector of reactor and effluent concentration (mg/L); |
| $\xi_{in}$ | vector of influent concentration (mg/L); |
| $\mu_H$ | specific growth rate for heterotrophic biomass ($d^{-1}$); |
| $\mu_{H,max}$ | maximum specific growth rate for heterotrophic biomass ($d^{-1}$); |
| $\rho(\xi)$ | vector of reaction kinetics (mg/(L d)); |
| $\rho_j$ | process rate (mg/(L d)); |
| $\Theta$ | hydraulic residence time, HRT (d); |
| $\nu_{ij}$ | stoichiometric coefficient; |
| $\eta_g$ | correction factor of $\mu_H$ under anoxic conditions; |
| $\eta_h$ | correction factor for hydrolysis under anoxic conditions. |

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
