# Peer review of "A Comprehensive View of the ASM1 Dynamic Model: Study on a Practical Case"

_water, doi:10.3390/w14071046_

Round 1

Reviewer 1 Report

The presented article is interesting and nice work a very few corrections are necessary for improving the presentation of the paper. I suggest this manuscript for major revision.

Comments

  1. Some of the highlights are meaningless; they should be revised as per the journal standard.
  2. The references are not properly formatted. They should not be numbered in the manuscript text.
  3. The introduction should include the state-of-the-art literature review as well as the novelty of this work and its distinction from other works.
  4. Abstract needs to be revised and add the main result of the review.
  5. All statements in this section are vague and need to be reframed.
  6. Please provide a detailed description of the experimental setup.
  7. Overall structural resemble and framing of sentences need to be revised to improve readability and match the journal standard. The entire manuscript needs language correction.

Author Response

1- Highlights have been rewritten, more conclusive. Thanks

2- References were formatted numbered in text in accordance with Water instructions to authors. Please check references style for Water or for example doi:10.3390/w14050763

3- Introduction has been deeply revised and changed. Latest articles in ASM1 have been better commented and reorganized to explain the state of the art in dynamic modelling by ASM1. Thanks for indication.

4- and 5- Abstract has been rewritten considering also the review in introduction. Thanks.

6- Experimental setup has been included in the new version of the manuscript: section 2.4. Thanks for indication.

7- It is true that the manuscript needed a complete language revision. The new version of the manuscript has been carefully revised and improved. Thanks.

Reviewer 2 Report

Figure 1. Is there any consideration for return sludge effects?

Table 4 should be removed in the main text because these data would be used for estimation for the values obtained from fine tuning.

Author Response

---- Figure 1 is a general figure for a complete mix reactor in which recycle has not been drawn. The objective of this figure is to show the significance of x, vector of concentration, which is assumed to S in stationary models. The figure has to be interpreted as an overall biological process that could include recycle. Input and output concentrations are shown, and reactor and settler are supposed to be included.

But recycle is not considered in ASM1, the mass balance formulation is for an in-out model.

That is a good point, for improving ASM1, thank you so much, but I think recycle has not to be included in the figure because is not distinguished in ASM1.

---- Table 4 has been placed in Supplementary material (Table 1S). This table was repetitive in the manuscript. Thanks.

English language, sentences and grammar in text have been carefully revised.

Reviewer 3 Report

1.This paper showed how simplified ASM1 model is applicable in one of practical operation of WWTP.

  1. It is difficult to find originality of this content.

 If the parameters in this model were assumed from independent experiments of this

process, the applicability of this model will be proved.

However, all data were taken from the process, it is a kind of curve-fitting of the model to the data.

  1. Many models are proposed in mathematical analysis, but all are just the application examples to specific cases.
  2. It is skeptical whether engineers in wastewater treatment plants appreciate this kind of research.

Author Response

1- and 2-  In the new version of the manuscript I tried to focus better the novelty and originality of the article (end of introduction and abstract). This is not a new model or a new process, this is an explanation of how to use and apply ASM1 to WWTP data, which from my point of view was not explained in detail before. Thanks.

3-  From my point of view this is a not very thoughtful sentence.

Dynamic mathematical modelling permits to simulate experimental data. Of course, parameters (constant variables) are characteristic of the specific case, but the model gives the values of kinetic parameters and COD. These are the identity signs of the biological treatment.

The objective of my work is not just the application example to a specific case, is to know the behaviour of the output COD in the WWTP and its variation because of environmental changes.

Thanks for comments and effort.

4-  I would like to rewrite which Morgens Henze did in 1987, in the original ASM1 (Introduction of the article):

Mathematical models are powerful tools by which the designers of biological wastewater systems can investigate the performance of a number of potential systems under a variety of conditions”.

Although this sentence was written 30 years ago, dynamic modelling in WWTPs permits to predict the output value of COD before happening, when environmental conditions change.

This is really important for engineers working in wastewater treatment plants.

Thank you, I appreciate an interesting discussion about modelling and experimental data.

Text has been carefully revised to improve language and grammar.

Round 2

Reviewer 1 Report

Aceepted

Reviewer 3 Report

The revised paper improved significantly.